# Following successful anti-leishmanial treatment, neutrophil counts, CD10 expression and phagocytic capacity remain reduced in visceral leishmaniasis patients co-infected with HIV

**Yegnasew Takele[1,2], Emebet Adem[2¤], Tadele Mulaw[2], Ingrid Müller[1], James Anthony Cotton[3], Pascale Kropf[1] \***

**1** Department of Infectious Disease, Imperial College London, London, United Kingdom, **2** Leishmaniasis Research and Treatment Centre, University of Gondar, Ethiopia, **3** Wellcome Sanger Institute, Wellcome Genome Campus, Hinxton, United Kingdom

¤ Current address: University of Greenwich at Medway, Kent, United Kingdom
\* p.kropf@imperial.ac.uk

## Abstract

Visceral leishmaniasis (VL) patients co-infected with HIV (VL/HIV patients) experience frequent treatment failures, VL relapses, opportunistic infections, and higher mortality. Their immune system remains profoundly suppressed after clinical cure and they maintain higher parasite load. This is in contrast with patients with VL alone (VL patients). Since neutrophils play a critical role in the control of *Leishmania* replication and the regulation of immune responses, we tested the hypothesis that neutrophil activation status and effector functions are fully restored in VL, but not in VL/HIV patients. Our results show the neutrophil counts and all activation markers and effector functions tested in our study were reduced at the time of diagnosis in VL and VL/HIV patients as compared to controls. CD62L, CD63, arginase 1 expression levels and reactive oxygen species production were restored at the end of treatment in both groups. However, neutrophil counts, CD10 expression and phagocytosis remained significantly lower throughout follow-up in VL/HIV patients; suggesting that dysregulated neutrophils contribute to the impaired host defence against pathogens in VL/HIV patients.

## Author summary

Visceral leishmaniasis (VL) is a neglected tropical disease that affects the most vulnerable people in low and middle-income countries. It is caused by *Leishmania* parasites. VL can be fatal, however there are efficient treatments to control this disease. In contrast, patients with visceral leishmaniasis (VL patients) who are co-infected with HIV (VL/HIV patients), they harbour a higher parasite load and suffer from treatment failure and frequent relapses. In this study, we analysed the effector functions of neutrophils, a type of

**Data Availability Statement:** All relevant data are within the manuscript and its Supporting Information files.

**Funding:** YT is funded by a Wellcome Trust Training Fellowship in Public Health and Tropical Medicine (204797/Z/16/Z). JAC is funded by Wellcome via core funding of the Wellcome Sanger Institute (grant 206194). The funders had no role in study design, data collection and analysis, decision to publish, or preparation of the manuscript.

**Competing interests:** The authors have declared that no competing interests exist.

blood cells that has the ability to take up *Leishmania* parasites and kill them. We show that over time in VL/HIV patients, the neutrophil counts remain significantly lower as compared to VL patients and healthy controls and they also display an impaired capacity to phagocytose. These results suggest that dysregulated neutrophils contribute to the impaired control of parasite replication in VL/HIV patients.

## Introduction

Visceral leishmaniasis (VL) is a neglected tropical disease, affecting the poorest populations. Worldwide, 78 countries are endemic for VL with 17,082 new cases reported in 2018 [1]. Due to the poor surveillance of this disease, it is generally accepted that this number is a substantial underestimate of the real burden of VL. This disease is caused by infections with parasites of the *Leishmania* (*L.) donovani* species complex. The parasites are transmitted to the hosts during the blood meal of sandfly vectors. In Ethiopia, VL is one of the most important vector-born disease, with >3 millions of individuals living in areas that are endemic for this disease [2]. Whereas most individuals infected with *L. donovani* will stay asymptomatic, a small percentage will develop VL. VL is characterised by fever, hepatosplenomegaly, wasting and pancytopenia; this stage of the disease is often fatal if not treated.

HIV co-infection of VL patients is one of the major challenges for the control of VL: HIV infection increases the risk of progression from asymptomatic *Leishmania* infection to VL and VL accelerates HIV disease progression. Ethiopia is the country is Africa with the highest rate of VL/HIV co-infections, with up to 23% of VL patients co-infected with HIV [3]. As compared to VL patients, VL/HIV patients experience more frequent treatment failures and VL relapses and higher mortality, as well as more opportunistic infections [4–6]. The mechanisms accounting for the increased rate of VL relapse in VL/HIV co-infected patients are poorly characterised. Markers such as low $CD4^+$ T cell counts, high parasite loads at the time of VL diagnosis and during follow-up, not being on ART at the time of VL diagnosis and antigenuria have shown variable degrees of predictive accuracy [7–10]. Another predictive marker of VL relapse in VL/HIV co-infected patients is a previous history of VL relapse [11], though this has led to conflicting results: a study by Abongomera *et al.* showed no association between a previous history of relapse and increased risks of future relapse [12]. We recently showed that in Northwest Ethiopia, 78.1% of VL/HIV patients experienced at least one episode of VL relapse over a period of three years. Importantly, we showed that despite clinical cure, VL/HIV patients maintained high parasite loads, failed to restore both $CD4^+$ T cell counts and antigen-specific IFNγ production and maintain higher expression of PD1 on $CD4^+$ and $CD8^+$ T cells. We also showed that in the plasma of VL/HIV patients, proinflammatory cytokines such as TNFα, IL-8 and IL-6 remained high throughout follow-up [13]. Neutrophils are amongst the first cells to be recruited to sites of inflammation, where they are one of the main mediators of inflammation. While this can sometimes result in tissue injury, neutrophils also play a key role in the resolution of inflammation [14–17]. In addition, neutrophils can eliminate pathogens via several mechanisms such as phagocytosis, production of toxic molecules such as reactive oxygen species (ROS), anti-bacterial proteins, and neutrophil extracellular traps (NETs) [14–17]. Interestingly, phagocytosis of different *Leishmania* species does not always result in the killing of the parasites [18]. *L. donovani* can be phagocytosed and killed by human neutrophils [19], but can survive in NETs *in vitro* [20]. A recent study showed that infection of human neutrophils with *L. donovani* resulted in increased glycolysis that seems to contribute to parasite survival, since inhibition of glycolysis resulted in lower parasite load [21]. Exposure to

ATP and UTP has been shown to enhance anti-*Leishmania* activity [21]. *L. donovani* can also induce autophagy in neutrophils, that in turn can facilitate their silent entry into macrophages [22]. In addition, *Leishmania*-containing phagosomes in neutrophils have been shown to avoid fusion with the granules involved in acidification of vacuoles, thereby contributing to parasite survival [23].

We have shown that in VL patients, a high proportion of neutrophils are immature, highly activated and display impaired effector functions [24]. A limited number of studies have analysed the role of neutrophils in the pathogenesis of human VL, but those performed with VL/HIV co-infected patients are particularly sparse; therefore, the role of neutrophils in the poor of clinical outcome or VL/HIV patients is still unclear. Here, our aim was to test the hypothesis that after successful anti-leishmanial treatment, neutrophil activation status and effector functions are fully restored over time in VL, but not in VL/HIV patients.

## Materials and methods

### Ethics statement

This study was approved by the Institutional Review Board of the University of Gondar (IRB, reference O/V/P/RCS/05/1572/2017), the National Research Ethics Review Committee (NRERC, reference 310/130/2018) and Imperial College Research Ethics Committee (ICREC 17SM480). Informed written consent was obtained from each patient and control.

### Patient recruitment

For this cross-sectional study, we used the three cohorts of individuals described in [13]; in brief, we recruited 25 healthy male non-endemic controls (median age 28.0 years (24.5–35.0)), 50 male patients with visceral leishmaniasis (VL patients, median age 25 years (20.0–29.8)) and 49 VL patients co-infected with HIV (VL/HIV patients, median age 33.5 years (28.0–38.8)) from the Leishmaniasis Research and Treatment Centre (LRTC), University of Gondar, Ethiopia. Patients age <18 years were excluded. VL and HIV diagnosis and treatments were all performed according to national guidelines in Ethiopia [25,26]. 46 VL/HIV patients were on antiretroviral therapy (ART) at the time of VL diagnosis, the remaining three started ART at the end of the anti-leishmanial treatment. At the end of treatment, all VL patients were clinically cured, as defined by the National Guidelines [25]: patients look improved, were afebrile, usually have a smaller spleen size and had an improved haematological profile. A test of cure was used for VL/HIV patients to decide if they could be discharged from hospital; a negative test of cure is defined in the National Guidelines [25] as follows: patients look improved, afebrile, and usually have a smaller spleen size, parasitological cure (absence of amastigotes in splenic aspirates) and an improved haematological profile. When the test of cure was still positive, treatment was continued until it became negative.

Anti-leishmanial treatments and duration, HIV treatments, number and time of VL relapses and clinical parameters have been described in [13].

VL and VL/HIV patients were recruited at four different time points: time of diagnosis (ToD); end of treatment (EoT); 3 months post the end of treatment (3m); and 6–12 months post the end of treatment (6-12m).

### Sample collection and processing

7.5ml of blood were collected and distributes as follows:

- 2.5ml in EDTA tubes for neutrophil counts.

- 5ml in heparin tubes for neutrophil isolation.

Neutrophil counts were measured as part of the complete blood cell counts using a Sysmex XP-300 automated haematology analyser, (USA) following the manufacturer's instruction. Eightcheck-3wp (USA) control samples were tested before running the patient samples.

Neutrophils were isolated as described in [27]. Briefly, following density gradient centrifugation on HistopaqueH-1077 (Sigma), the supernatant was discarded, and the pellet (containing erythrocytes and neutrophils) was resuspended in PBS; 3% dextran sulfate was added, and the tube was left to stand for 20min at room temperature to allow for erythrocyte sedimentation. The supernatant was collected, washed, and suspended in red cell lysis buffer and incubated at 4˚C for 15min. The suspension was washed with PBS and the neutrophil sediment was resuspended in PBS.

## Flowcytometry

The following antibodies were used: anti-human CD62L$^{PE/Cy7}$ (DREG-56), anti-human CD15$^{APC}$ (H198), anti-human CD10$^{FITC}$ (eBioSN5c), anti-human CD63$^{PE/Cy7}$ (H5C6) (eBiosciences) and anti-human arginase 1$^{PE}$ (14D2C43) (BioLegend).

For intracellular staining of arginase 1, cells were permeabilized with 0.25% saponin for 15min at room temperature in the dark and washed with PBS. Anti-arginase-1 was added and incubated for 15min. Cells were then washed with PBS, resuspended in 300ul of PBS and analysed immediately on the flow cytometer.

## Phagocytosis of *E. coli* bioparticles

Uptake of pHrodo Green *Escherichia coli* BioParticles Conjugate (Molecular probe) was used to assess the phagocytic capacity of neutrophils, as described in the manufacturer's protocol. Purified neutrophils were re-suspended in 10% RPMI at a concentration $1x10^5$ cells and *E. coli* particles were added at a 1:10 ratio. Cells were then incubated at 37˚C for 1hour. At the end of incubation cells were washed, resuspended in PBS and analysed immediately on the flow cytometer.

## Reactive oxygen species

Total ROS detection kit (Enzo Life Sciences, Inc.) was used to evaluate the production of ROS by neutrophils, as described in the manufacturer's protocol. $1x10^5$ neutrophils were incubated in 0.1% complete RPMI at 37˚C, 5% $CO_2$ in the presence and absence of 200μM of pyocyanin (PYO) for 60min. Cells were then washed in PBS at 1200rpm, 500μl of detection reagent was added and the cells were incubated at 37˚C, 5% $CO_2$ for 30min, CD15 was added during the last 15min of incubation and at the end of the incubation, the cells were analysed immediately on the flow cytometer.

Acquisition was performed by using a BD Accuri C6 flow cytometer (BD Biosciences) and data analysed using BD Accuri C6 analysis software version 1.0.264.21.2.6.2.3.

## Statistical analysis

Data were evaluated for statistical differences as specified in the legend of each figure. The following tests were used: Kruskal-Wallis, Mann-Whitney and Wilcoxon matched-pairs signed rank tests. Differences were considered statistically significant at $p<0.05$. Results are expressed as median with interquartile range. $^*$ = p<0.05, $^{**}$ = p<0.01, $^{***}$ = p<0.001 and $^{****}$ = p<0.0001.

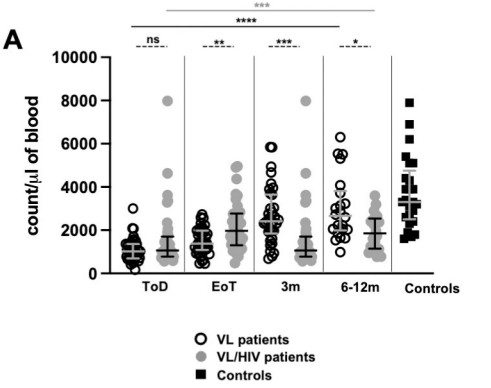
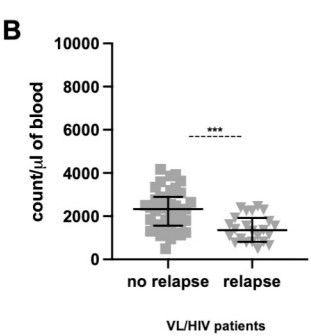

**Fig 1. Comparison of neutrophil counts between VL and VL/HIV patients and controls over time.** Neutrophil counts were measured in the whole blood of controls (n = 25), VL (ToD: n = 41, EoT: n = 42; 3m: n = 34, 6-12m: n = 22) and VL/HIV (ToD: n = 37, EoT: n = 36, 3m: n = 37, 6-12m: n = 20) patients as described in Material and Methods. **A.** Comparison of neutrophil counts over time in VL, VL/HIV and controls. **B.** Comparison of neutrophil counts from VL/HIV patients who did not relapse (n = 39) and those who relapsed (n = 23) after successful anti-leishmanial treatment, during the 3m and 6–12 follow-up period. If a patient did not relapse during the two-time points of follow-up and if a patient relapsed at both 3 and 6–12 months, this is represented as 2 measurements. Each symbol represents the value for one individual. Results are presented as median with interquartile range. Statistical differences between VL and VL/HIV patients at each time point or between VL/HIV patients who did not relapse and those who relapsed were determined using a Mann-Whitney test and statistical differences between the 4 different time points for each cohort of patients were determined by Kruskal-Wallis test. ToD = Time of Diagnosis; EoT = End of Treatment; 3m = 3 months post EoT; 6-12m = 6–12 months post EoT. ns = not significant.

## Results

We first compared neutrophil counts in the blood of VL and VL/HIV patients over time. As shown in Fig 1A and Table 1, neutrophil counts were significantly lower in both groups as compared to controls at ToD. In VL patients, these counts stayed similar at EoT, but increased significantly at 3m and were restored 6-12m post the end of treatment (Fig 1A). In VL/HIV patients, despite an increase in neutrophil counts at EoT, these numbers stayed significantly lower until the end of follow-up. We have previously shown that following successful clinical cure, no VL patients experienced VL relapse, but that 78.1% of VL/HIV patients relapse at least once [13]. We therefore measured neutrophil counts in VL/HIV patients who relapsed during follow-up and compared them with those who did not; our results show a significantly

**Table 1. Comparison of neutrophil counts between VL and VL/HIV patients and controls over time.**

| VL ToD | Controls | *p* value | VL/HIV ToD | Controls | *p* value |
|---|---|---|---|---|---|
| 1020 (697–1359) | | <0.0001 | 1059 (780–1706) | | <0.0001 |
| **VL EoT** | 3300 (2500–4750) | | **VL/HIV EoT** | 3300 (2500–4750) | |
| 1376 (1089–1976) | | <0.0001 | 1972 (1307–2772) | | <0.0001 |
| **VL 3m** | | | **VL/HIV 3m** | | |
| 2409 (1850–3649) | | 0.0098 | 1059 (780–1706) | | <0.0001 |
| **VL 6-12m** | | | **VL/HIV 6-12m** | | |
| 2659 (1988–3808) | | 0.0933 | 1853 (1149–2541) | | <0.0001 |

Neutrophils were counted in the whole blood of controls (n = 25), VL (ToD: n = 41, EoT: n = 42; 3m: n = 34, 6-12m: n = 22) and VL/HIV (ToD: n = 37, EoT: n = 36, 3m: n = 37, 6-12m: n = 20) patients as described in Materials and Methods.

Statistical differences between VL at each time point during follow-up and controls; and between VL/HIV and controls at each time point during follow-up and controls were determined using a Mann-Whitney test. Results are presented as median with interquartile range.

ToD = Time of Diagnosis; EoT = End of Treatment; 3m = 3 months post EoT; 6-12m = 6–12 months post EoT.

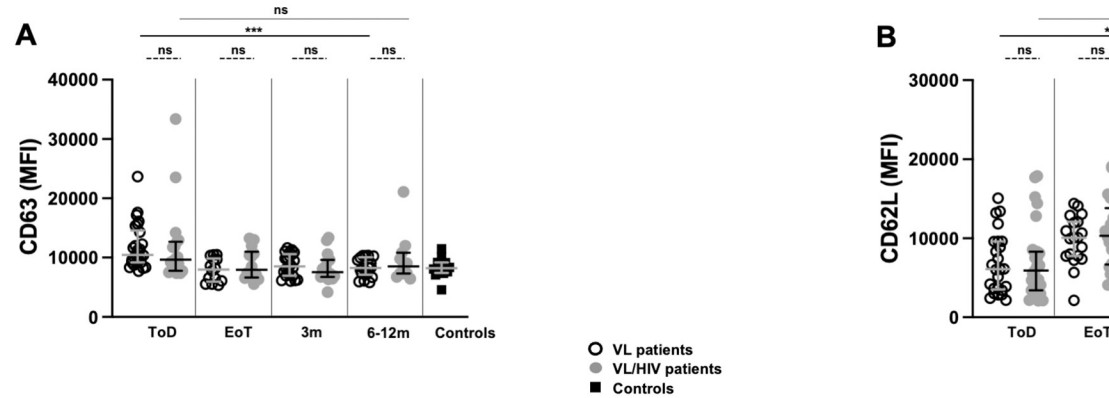

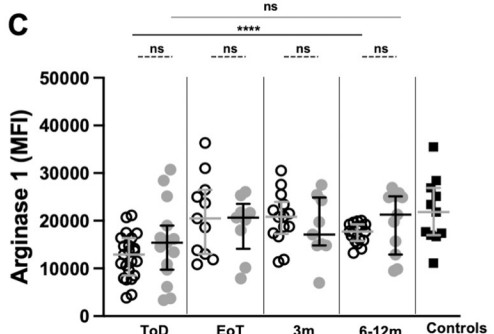

**Fig 2. Comparison of neutrophil CD63, CD62L and arginase 1 expression levels between VL and VL/HIV patients and controls over time.** Neutrophils were isolated from whole blood by dextran sedimentation as described ibn Materials and Methods: **A.** The expression levels (MFI = Median Fluorescence Intensity) of CD63 expression levels were measured on neutrophils from controls (n = 12), VL (ToD: n = 26, EoT: n = 14 3m: n = 22, 6-12m: n = 18) and VL/HIV (ToD: n = 18, EoT: n = 14, 3m: n = 15, 6-12m: n = 11) patients by flow cytometry as described in Materials and Methods. **B.** The expression levels (MFI) of CD62L expression levels were measured on neutrophils from controls (n = 18), VL (ToD: n = 25, EoT: n = 19 3m: n = 24, 6-12m: n = 34) and VL/HIV (ToD: n = 27, EoT: n = 25, 3m: n = 18, 6-12m: n = 23) patients by flow cytometry as described in Materials and Methods; **C.** The intracellular expression levels (MFI) of Arginase 1 levels were measured in neutrophils from controls (n = 11), VL (ToD: n = 23, EoT: n = 11, 3m: n = 14, 6-12m: n = 17) and VL/HIV (ToD: n = 15, EoT: n = 9, 3m: n = 9, 6-12m: n = 11) patients by flow cytometry as described in Materials and Methods. Each symbol represents the value for one individual. Results are presented as median with interquartile range. Statistical differences between VL and VL/HIV patients at each time point were determined using a Mann-Whitney test and statistical differences between the 4 different time points for each cohort of patients were determined by Kruskal-Wallis test. ToD = Time of Diagnosis; EoT = End of Treatment; 3m = 3 months post EoT; 6-12m = 6–12 months post EoT. ns = not significant.

higher number of neutrophils in the latter group (Fig 1B). However, the median neutrophil count of VL/HIV patients who did not relapse (2330 neutrophils/μl of blood [1560–2895]) was still significantly lower ($p = 0.0001$) as compared to controls (3300 neutrophils/μl of blood [2500–4750]).

Next we evaluated neutrophil activation status in VL and VL/HIV patients over time. Results presented in Fig 2A show that the levels of CD63, a molecule that becomes upregulated on neutrophil cell surface following activation [28], were higher on neutrophils from VL and VL/HIV as compared to controls ($p = 0.0005$ and $p = 0.0310$, respectively, Table 2) at ToD. CD63 Median Fluorescence Intensity (MFI) decreased at EoT in both groups, although not significantly in the VL/HIV patients and were restored to levels similar to those of control (Fig 2A, Table 2). There were no significant differences ($p = 0.7473$) in CD63 MFI between VL/HIV patients (7345 [6899–9962]) who relapsed and those who didn't (8354 [6683–10052]).

Next we measured the expression levels of CD62L, a ligand that is shed from the neutrophil cell surface following activation [29,30]. CD62L MFI were lower on neutrophils from VL and

**Table 2. Comparison of neutrophil CD63 expression levels between VL and VL/HIV patients and controls over time.**

| VL ToD | Controls | p value | VL/HIV ToD | Controls | p value |
|---|---|---|---|---|---|
| 10448 (9180–14582) | 8222 (7405–9175) | 0.0005 | 9648 (7776–12689) | 8222 (7405–9175) | 0.0310 |
| **VL EoT** | | | **VL/HIV EoT** | | |
| 7991 (5958–9856) | | 0.7424 | 7943 (6640–10991) | | 0.8995 |
| **VL 3m** | | | **VL/HIV 3m** | | |
| 8533 (6936–10597) | | 0.6633 | 7562 (6765–9602) | | 0.5720 |
| **VL 6-12m** | | | **VL/HIV 6-12m** | | |
| 8264 (7115–10040) | | 0.9751 | 8510 (7319–10821) | | 0.8202 |

Neutrophils were isolated from whole blood by dextran sedimentation and the expression levels (MFI = Median Fluorescence Intensity) of CD63 on neutrophils from controls (n = 12), VL (ToD: n = 26, EoT: n = 14 3m: n = 22, 6-12m: n = 18) and VL/HIV (ToD: n = 18, EoT: n = 14, 3m: n = 15, 6-12m: n = 11) patients were measured by flow cytometry as described in Materials and Methods. Statistical differences between VL patients at each time point during follow-up and controls; and between VL/HIV patients and controls at each time point during follow-up and controls were determined using a Mann-Whitney test. Results are presented as median with interquartile range.

ToD = Time of Diagnosis; EoT = End of Treatment; 3m = 3 months post EoT; 6-12m = 6–12 months post EoT.

VL/HIV patients as compared to controls (Fig 2B; $p$ = 0.0003 and $p$ = 0.0016, respectively, Table 3) at ToD; CD62L MFI were restored to levels similar to those of controls at EoT in both groups (Fig 2B, Table 3). There were no significant differences ($p$ = 0.1750) in CD62L MFI between VL/HIV patients who relapsed (13380 [7986–17068]) and those who didn't (10022 [7155–13582]).

Arginase 1 is an enzyme present in the primary granules of neutrophils that has immuno-modulatory properties [24,31]. Here we show that the levels of arginase 1 were significantly lower in both VL and VL/HIV patients at ToD as compared to controls ($p$<0.0001 and $p$ = 0.0196, respectively, Fig 2C, Table 4). These levels increased after treatment, however not significantly in the VL/HIV group, and were restored to levels similar to those of controls at EoT in both groups (Fig 2C, Table 4). There were no significant differences ($p$ = 0.7959) in arginase 1 MFI between VL/HIV patients who relapsed (19755 [14817–25017]) and those who didn't (21286 [12894–26922]).

We also measured the levels of CD10, a molecule present only on segmented neutrophils [29], that can regulate both inflammatory processes [32] and T cell effector functions [33].

**Table 3. Comparison of neutrophil CD62L expression levels between VL and VL/HIV patients and controls over time.**

| VL ToD | Controls | p value | VL/HIV ToD | Controls | p value |
|---|---|---|---|---|---|
| 6091 (3525–9594) | 12229 (7368–14764) | 0.0003 | 5924 (3410–8275) | 12229 (7368–14764) | 0.0016 |
| **VL EoT** | | | **VL/HIV EoT** | | |
| 10065 (7736–12086) | | 0.1594 | 10310 (6667–13813) | | 0.4136 |
| **VL 3m** | | | **VL/HIV 3m** | | |
| 9133 (5459–11383) | | 0.0307 | 11639 (7420–14633) | | 0.7366 |
| **VL 6-12m** | | | **VL/HIV 6-12m** | | |
| 10802 (9547–14043) | | 0.8084 | 13108 (7526–15208) | | 0.9840 |

Neutrophils were isolated from whole blood by dextran sedimentation and the expression levels (MFI = Median Fluorescence Intensity) of CD62L on neutrophils from controls (n = 18), VL (ToD: n = 25, EoT: n = 19 3m: n = 24, 6-12m: n = 34) and VL/HIV (ToD: n = 27, EoT: n = 25, 3m: n = 18, 6-12m: n = 23) patients were measured by flow cytometry as described in Materials and Methods. Statistical differences between VL patients at each time point during follow-up and controls; and between VL/HIV patients and controls at each time point during follow-up and controls were determined using a Mann-Whitney test. Results are presented as median with interquartile range.

ToD = Time of Diagnosis; EoT = End of Treatment; 3m = 3 months post EoT; 6-12m = 6–12 months post EoT.

**Table 4. Comparison of neutrophil Arginase 1 expression levels between VL and VL/HIV patients and controls over time.**

| VL ToD | Controls | *p* value | VL/HIV ToD | Controls | *p* value |
|---|---|---|---|---|---|
| 12904 (8516–16373) | 21799 (16978–26909) | <0.0001 | 15403 (9729–18996) | 21799 (16978–26909) | 0.0196 |
| **VL EoT** | | | **VL/HIV EoT** | | |
| 20482 (13031–26486) | | 0.7969 | 20664 (14084–23523) | | 0.5516 |
| **VL 3m** | | | **VL/HIV 3m** | | |
| 20830 (17236–23928) | | 0.7777 | 17096 (14817–24848) | | 0.2857 |
| **VL 6-12m** | | | **VL/HIV 6-12m** | | |
| 17729 (15791–19232) | | 0.0978 | 21286 (12894–25108) | | 0.5292 |

Neutrophils were isolated from whole blood by dextran sedimentation and the expression levels (MFI = Median Fluorescence Intensity) of arginase 1 in neutrophils controls (n = 11), VL (ToD: n = 23, EoT: n = 11, 3m: n = 14, 6-12m: n = 17) and VL/HIV (ToD: n = 15, EoT: n = 9, 3m: n = 9, 6-12m: n = 11) patients were measured by flowcytometry as described in Materials and Methods. Statistical differences between VL patients at each time point during follow-up and controls; and between VL/HIV patients and controls at each time point during follow-up and controls were determined using a Mann-Whitney test. Results are presented as median with interquartile range.

ToD = Time of Diagnosis; EoT = End of Treatment; 3m = 3 months post EoT; 6-12m = 6–12 months post EoT.

Results presented in Fig 3A show that CD10 is significantly lower in both groups at ToD as compared to controls (*p*<0.0001, Table 5). CD10 expression increased at EoT in VL patients and was restored at 3m (Fig 3A, Table 5). In contrast, in VL/HIV patients, CD10 levels remained similar at EoT, and only increased marginally over time. CD10 expression on neutrophils was also significantly higher in VL/HIV patients who did not relapse over time (Fig 3B). But still remained significantly lower (*p* = 0.0406) as compared to controls (15887 [13445–19799] and (14653 [11599–16298], respectively).

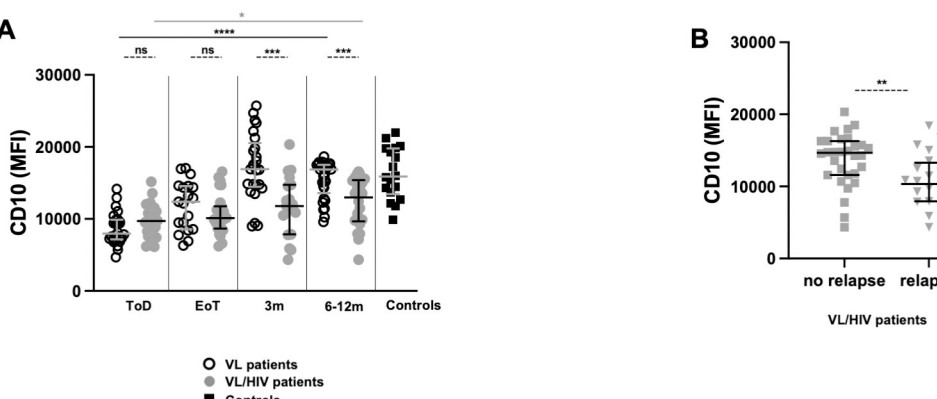

**Fig 3. Comparison of neutrophil CD10 expression levels between VL and VL/HIV patients and controls over time.** Neutrophils were isolated from whole blood by dextran sedimentation and the expression levels (MFI = Median Fluorescence Intensity) of CD10 were measured by flow cytometry on neutrophils from controls (n = 18), VL (ToD: n = 26, EoT: n = 19, 3m: n = 25, 6-12m: n = 29) and VL/HIV (ToD: n = 27, EoT: n = 26, 3m: n = 19, 6-12m: n = 22) patients as described in Materials and Methods. **A.** Comparison of CD10 MFI on neutrophils from VL, VL/HIV and controls patients over time. **B**. Comparison of CD10 MFI on neutrophils from VL/HIV patients who did not relapse (n = 30) and those who relapsed (n = 20) after successful anti-leishmanial treatment, during the 3m and 6–12 follow-up period. If a patient did not relapse during the two-time points of follow-up and if a patient relapsed at both 3 and 6–12 months, this is represented as 2 measurements. Each symbol represents the value for one individual. Results are presented as median with interquartile range. Statistical differences between VL and VL/HIV patients at each time point or between VL/HIV patients who did not relapse and those who relapsed were determined using a Mann-Whitney test and statistical differences between the 4 different time points for each cohort of patients were determined by Kruskal-Wallis test. ToD = Time of Diagnosis; EoT = End of Treatment; 3m = 3 months post EoT; 6-12m = 6–12 months post EoT. ns = not significant.

**Table 5. Comparison of neutrophil CD10 expression level between VL and VL/HIV patients and controls over time.**

| VL ToD | Controls | *p* value | VL/HIV ToD | Controls | *p* value |
|---|---|---|---|---|---|
| 7974 (7159–9886) | 15887 (13445–19799) | <0.0001 | 9707 (8215–11027) | 15887 (13445–19799) | <0.0001 |
| **VL EoT** | | | **VL/HIV EoT** | | |
| 12391 (8568–14561) | | 0.0016 | 10121 (8669–11771) | | <0.0001 |
| **VL 3m** | | | **VL/HIV 3m** | | |
| 16914 (14455–20527) | | 0.5341 | 11792 (7874–14741) | | 0.0023 |
| **VL 6-12m** | | | **VL/HIV 6-12m** | | |
| 16866 (13597–17523) | | 0.4641 | 12996 (9655–15384) | | 0.0016 |

Neutrophils were isolated from whole blood by dextran sedimentation and the expression levels (MFI = Median Fluorescence Intensity) of CD10 on neutrophils from controls (n = 18), VL (ToD: n = 26, EoT: n = 19, 3m: n = 25, 6-12m: n = 29) and VL/HIV (ToD: n = 27, EoT: n = 26, 3m: n = 19, 6-12m: n = 22) patients were measured by flowcytometry as described in Materials and Methods. Statistical differences between VL patients at each time point during follow-up and controls; and between VL/HIV patients and controls at each time point during follow-up and controls were determined using a Mann-Whitney test. Results are presented as median with interquartile range.

ToD = Time of Diagnosis; EoT = End of Treatment; 3m = 3 months post EoT; 6-12m = 6–12 months post EoT.

Next we assessed two effector functions of neutrophils, ROS production and phagocytosis. As shown in S1 Fig, the median ROS production by neutrophils from both groups of patients was significantly increased in response to pyocyanin at all time points. Results in Fig 4 show that the capacity of neutrophils to produce ROS in response to pyocyanin was significantly impaired at ToD in VL and VL/HIV patients as compared to controls (*p* = 0.0008 and *p* = 0.0011, respectively, Table 6). It remained significantly lower as compared to controls in VL patients at EoT and 3m but was restored at 6-12m (Table 6). ROS production was already restored at EoT (Fig 4, Table 6) in VL/HIV patients. The ability of neutrophils to produce ROS was similar (*p* = 0.1777) in VL/HIV patients who relapse (206464 [119571]) and those who didn't relapse during follow-up (227781 [155669–259241]).

Next we measured the capacity of neutrophils to phagocytose *E. coli* bioparticles. As shown in Fig 5, at ToD phagocytosis was significantly impaired in VL and VL/HIV patients (*p* = 0.0006 and *p* = 0.0003, respectively, Table 7). This was restored to levels similar to those of controls at EoT in VL patients, but remained impaired at all time points in VL/HIV patients (Table 7). In the latter group, the ability of neutrophils to phagocytose bioparticle was similarly impaired in patients who relapse and those who didn't during follow-up (Fig 5B, Table 7).

## Discussion

Neutrophils are the most abundant cells in the blood and play major roles in several aspects of the immune response. They are rapidly recruited to the site of injury and pathology where they can kill pathogens via different mechanisms such as phagocytosis, production of ROS and NET formation [18]. They can also exert immunosuppressive activities [27,33,34]. And they play a major role in the induction as well as the resolution of inflammation [14–17].

HIV infected individuals often experienced neutropenia, this can be reversed by ART [35]. Neutrophils from HIV positive individuals also display impaired effector functions, such as phagocytosis, oxidative burst and chemotaxis [36]. These dysfunctions have been associated with a higher risk of opportunistic infections. Some of these effector functions are not fully restored with ART treatment (reviewed in [37]).

Neutrophils have been shown to play both a protective and a detrimental role in experimental models of leishmaniases [38]. Their role in human visceral leishmaniasis is not fully understood and has been poorly characterized in VL patients co-infected with HIV. Our previous

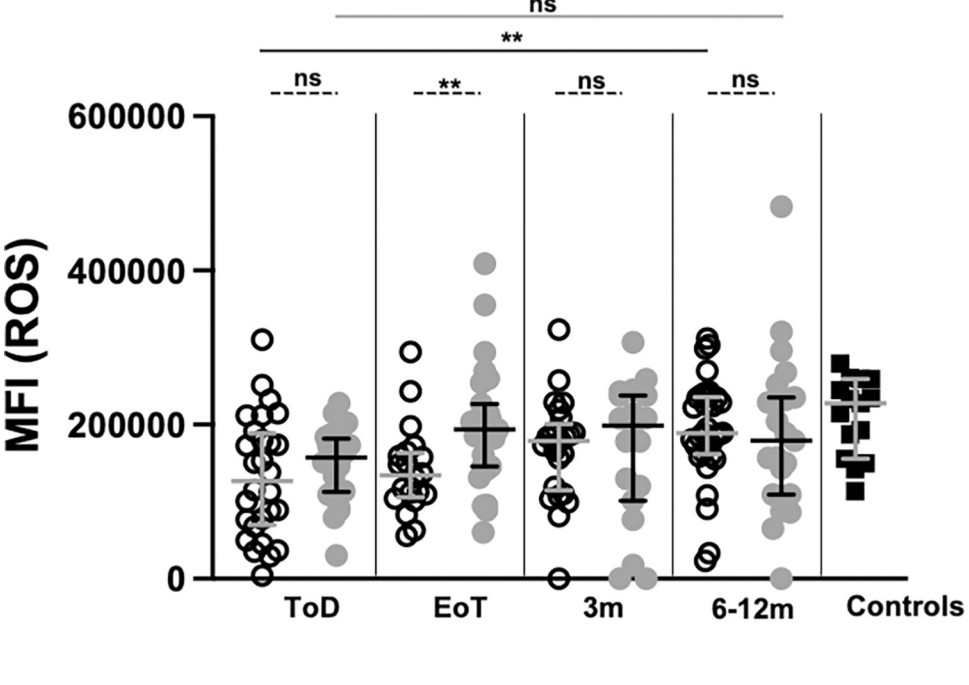

**Fig 4. Comparison of ROS production by neutrophils between VL and VL/HIV patients and controls over time.**
Neutrophils were isolated by dextran sedimentation from the blood of controls (n = 15), VL (ToD: n = 28, EoT: n = 20, 3m: n = 25, 6-12m: n = 33) and VL/HIV (ToD: n = 22, EoT: n = 27, 3m: n = 19, 6-12m: n = 23) patients and incubated in the presence or absence of pyocyanin. The production of ROS was evaluated by subtracting the MFI obtained from the neutrophils incubated without pyocyanin (baseline) from those incubated in the presence of pyocyanin, as measured by flow cytometry. Each symbol represents the value for one individual. Results are presented as median with interquartile range. Statistical differences between VL and VL/HIV patients at each time point were determined using a Mann-Whitney test and statistical differences between the 4 different time points for each cohort of patients were determined by Kruskal-Wallis test. ToD = Time of Diagnosis; EoT = End of Treatment; 3m = 3 months post EoT; 6-12m = 6–12 months post EoT. ns = not significant.

study [24] has shown that in VL patients, neutrophil effector functions and activation status were impaired at time of VL diagnosis, and that these function and activation markers were not always fully restored at the end of treatment. We proposed that dysfunctional neutrophils play a crucial role in the uncontrolled parasite replication and in the systemic inflammatory response in VL patients at the time of diagnosis. Here we extended these findings and showed that in patients infected with *L. donovani* alone, the activation status, as measured by the expression levels of CD63, CD62L and arginase 1 as well as their capacity to phagocytose bio-particles and produce ROS in response to pyocyanin is restored over time. We also show that in VL patients, the median neutrophil counts remain lower as compared to controls until 3m post EoT and are restored at 6-12m. In contrast, in VL/HIV patients, the median neutrophil counts remain significantly lower throughout follow-up and at 6-12m, were still 48% lower as compared to healthy controls. This neutropenia might be explained by several mechanisms:

i. Splenic sequestration, where blood cells are sequestered in the spleen of patients [39,40].

**Table 6. Comparison of ROS production by neutrophils between VL and VL/HIV patients and controls over time.**

| VL ToD | Controls | p value | VL/HIV ToD | Controls | p value |
|---|---|---|---|---|---|
| 126879 (69968–188528) | 227781 (155644–259241) | 0.0008 | 157012 (112542–181547) | 227781 (155644–259241) | 0.0011 |
| **VL EoT** | | | **VL/HIV EoT** | | |
| 134333 (105756–163329) | | 0.0011 | 193855 (145505–226971) | | 0.2624 |
| **VL 3m** | | | **VL/HIV 3m** | | |
| 178722 (114847–200766) | | 0.0241 | 198648 (101379–237668) | | 0.1543 |
| **VL 6-12m** | | | **VL/HIV 6-12m** | | |
| 188698 (161849–235517) | | 0.4022 | 179302 (109144–235073) | | 0.2326 |

Neutrophils were isolated by dextran sulfate sedimentation from the blood of controls (n = 15), VL (ToD: n = 28, EoT: n = 20, 3m: n = 25, 6-12m: n = 33) and VL/HIV (ToD: n = 22, EoT: n = 27, 3m: n = 19, 6-12m: n = 23) patients and incubated in the presence or absence of pyocyanin. The production of ROS was evaluated by substracting the MFI obtained from the neutrophils incubated without pyocyanin (baseline) from those incubated in the presence of pyocyanin, as measured by flowcytometry as described in Materials and Methods. Statistical differences between VL patients at each time point during follow-up and controls; and between VL/HIV patients and controls at each time point during follow-up and controls were determined using a Mann-Whitney test. Results are presented as median with interquartile range.

ToD = Time of Diagnosis; EoT = End of Treatment; 3m = 3 months post EoT; 6-12m = 6–12 months post EoT.

ii. Bone marrow failure: *Leishmania* and HIV can infect hematopoietic stem/progenitor cells [41–43] and thus impact haematopoiesis [42,44], this will contribute to neutropenia. And indeed, both HIV and *Leishmania* infections are associated with ineffective haematopoiesis and severe neutropenia [24,26,45]. Neutropenia in VL patients was shown to be associated with a low neutrophil reserve in the bone marrow [46]. HIV proteins have also been shown to have the ability to suppress CFU-granulocyte macrophage [47]. This will also contribute to neutropenia.

iii. Apoptosis: neutrophils will undergo apoptosis so that inflammation can be regulated. We and others have shown high levels of inflammatory cytokines in the plasma of VL patients [13,24,48] and VL/HIV patients [13]. It is therefore possible that the neutropenia characteristic of these patients results at least in part from excessive neutrophil apoptosis [49].

Neutropenia in VL patients can be reversed by injection of GM-CSF, this has been shown to be associated with a significant decrease of secondary infections [50]. This might explain the higher rate of opportunistic infections characteristic of these patients [6]. The poor prognosis of VL/HIV patients as compared to VL is likely to be further worsened by the impaired ability of neutrophils from VL/HIV patients to phagocytose bacterial particles throughout follow-up. This is likely to impede the effective control of opportunistic infections. And indeed, sepsis as well as opportunistic infections such as tuberculosis have been shown to be a predictor of death or poor treatment outcome in VL/HIV patients [6,51]. Whereas ROS production in response to pyocyanin was impaired at ToD in both VL and VL/HIV patients, it was restored faster in VL/HIV patients. This increased ability to produce ROS could be detrimental, as it could contribute to neutrophil apoptosis, as well as to sustained inflammation and tissue damage [49]. Whereas ROS can play a damaging effect, it has also been shown to play a positive role in the control of *Leishmania* parasites, as inhibition of ROS resulted in increased parasite survival [52,53].

CD63, CD62L and arginase 1 were all altered at time of diagnosis, suggesting that neutrophils are activated: CD63 is expressed in the primary granules of neutrophils and following activation of neutrophils, it becomes upregulated at their surface during degranulation [24,28]. CD62L is known to be shed from the neutrophil cell surface following activation [24,29,30]; arginase 1, an enzyme also found in the primary granules of neutrophils, is released in the

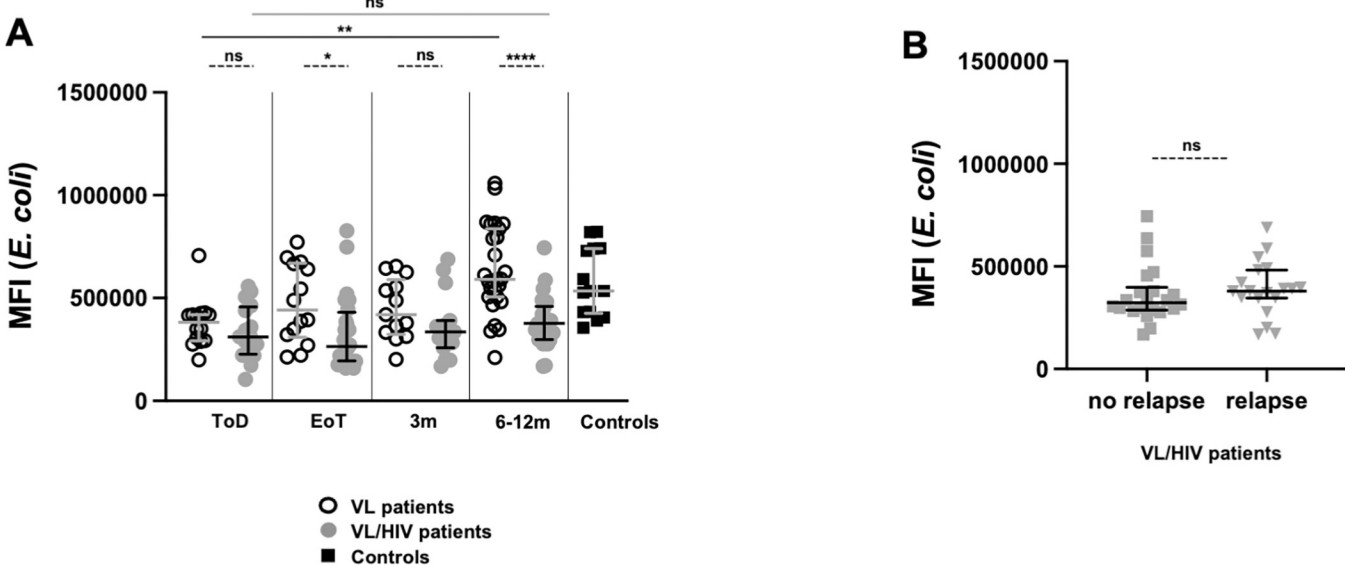

**Fig 5. Comparison of bioparticle phagocytosis between VL and VL/HIV patients and controls over time.** Neutrophils were isolated by dextran sedimentation from the blood of controls (n = 15), VL (ToD: n = 14, EoT: n = 14, 3m: n = 13, 6-12m: n = 27) and VL/HIV (ToD: n = 16, EoT: n = 24, 3m: n = 15, 6-12m: n = 26) patient and the MFI of *E. coli* particles were measured by flow cytometry, as described in Materials and Methods. Each symbol represents the value for one individual. **A.** Comparison of bioparticle phagocytosis by neutrophils from VL, VL/HIV and controls patients over time. **B**. Comparison of bioparticle phagocytosis by neutrophils from VL/HIV patients who did not relapse (n = 22) and those who relapsed (n = 19) after successful anti-leishmanial treatment, during the 3m and 6–12 follow-up period. If a patient did not relapse during the two-time points of follow-up and if a patient relapsed at both 3 and 6–12 months, this is represented as 2 measurements. Results are presented as median with interquartile range. Statistical differences between VL and VL/HIV patients at each time point or between VL/HIV patients who did not relapse and those who relapsed were determined using a Mann-Whitney test and statistical differences between the 4 different time points for each cohort of patients were determined by Kruskal-Wallis test. ToD = Time of Diagnosis; EoT = End of Treatment; 3m = 3 months post EoT; 6-12m = 6–12 months post EoT. ns = not significant.

microenvironment following degranulation of neutrophils [31]. These markers were restored at the end of treatment in both groups of patients, suggesting that these markers are not associated with neutrophil dysfunction in VL/HIV patients. In contrast, whereas CD10 was significantly lower on neutrophils of VL and VL/HIV patients at ToD, it was restored at 3m in VL patients, but remained lower in VL/HIV patients throughout follow-up. CD10 is expressed on mature segmented neutrophils. In our previous study [24], we showed that the lower expression of CD10 was accompanied by a significant increase in band neutrophils, suggesting that neutrophils in these patients were immature. Whereas in the current study, we did not look at smears of purified neutrophils to count the number of band and segmented neutrophils, it is plausible that there is a higher proportion of immature neutrophils in VL/HIV patients. The ineffective haematopoiesis [42,44] could result in the release of immature neutrophils in the circulation, that would at least partially explain the lower levels of CD10 on peripheral neutrophils in VL/HIV patients. CD10 is a neutral endopeptidase that can play an important role in the regulation of inflammatory processes. It is present on the membrane of the neutrophilic secretory granules [54] and its expression increases following activation of neutrophils [32]. CD10 can hydrolyse a variety of peptides that can promote inflammation [32,55]. It is therefore possible that the lack of CD10 expression on neutrophils contributes to the uncontrolled inflammation that is characteristic of VL/HIV patients over time [13]. Indeed, we have recently shown that VL/HIV patients have high levels of inflammatory cytokines in their plasma over time [13].

In addition to its role in inflammation, CD10 has been shown to play a role in the morphology and migration of neutrophils, as well as in some adhesion molecules [32]. Furthermore,

**Table 7. Comparison of phagocytosis between VL and VL/HIV patients and controls over time.**

| VL ToD | Controls | p value | VL/HIV ToD | Controls | p value |
|---|---|---|---|---|---|
| 381400 (292204–419881) | 534610 (425748–740569) | 0.0006 | 310407 (226407–457414) | 534610 (425748–740569) | 0.0003 |
| **VL EoT** | | | **VL/HIV EoT** | | |
| 441384 (309268–669432) | | 0.1251 | 264949 (194051–430165) | | 0.0001 |
| **VL 3m** | | | **VL/HIV 3m** | | |
| 419059 (323195–588931) | | 0.0683 | 336019 (258036–391006) | | 0.0006 |
| **VL 6-12m** | | | **VL/HIV 6-12m** | | |
| 590345 (505457–836869) | | 0.3198 | 376258 (297566–459460) | | 0.0002 |

Neutrophils were isolated by dextran sedimentation from the blood of controls (n = 15), VL (ToD: n = 14, EoT: n = 14, 3m: n = 13, 6-12m: n = 27) and VL/HIV (ToD: n = 16, EoT: n = 24, 3m: n = 15, 6-12m: n = 26) patient and the MFI of *E. coli* particles were measured by flow cytometry as described in Materials and Methods. Each symbol represents the value for one individual, the straight lines represent the median. Statistical differences between VL patients at each time point during follow-up and controls; and between VL/HIV patients and controls at each time point during follow-up and controls were determined using a Mann-Whitney test. Results are presented as median with interquartile range.

ToD = Time of Diagnosis; EoT = End of Treatment; 3m = 3 months post EoT; 6-12m = 6–12 months post EoT.

CD10 negative neutrophils have been shown to increase production of IFN-γ by CD4⁺ T cells [56].

As described in our previous publication [13], VL/HIV patients were treated with different combinational therapies. We cannot exclude that these treatments might have impacted on neutrophils. Indeed, AZT, a drug that was used in our cohort of VL/HIV patients, has been shown to result in neutropenia [57]. Furthermore, different classes of antiretroviral agents have been shown to impact on neutrophil effector functions, such as phagocytosis and ROS production, as well as on the expression levels of cell surface markers (summarised in [58]). However, due to the small numbers of patients receiving some of the treatments, it was not possible to dissect the impact of the different ART regimens on neutrophils.

In summary, our results show that in VL/HIV patients, neutrophil counts, as well as the ability of neutrophils to phagocytose bioparticle and express CD10, are not restored following treatment; suggesting that neutrophils play a key role in the poor prognosis of these patients.

## Supporting information

**S1 Fig. Comparison of ROS induction by pyocyanin in neutrophils between VL and VL/HIV patients and controls over time.** Neutrophils were isolated by dextran sedimentation from the blood of VL (ToD: n = 28, EoT: n = 20, 3m: n = 25, 6-12m: n = 33) and VL/HIV (ToD: n = 22, EoT: n = 27, 3m: n = 19, 6-12m: n = 23) patients and incubated in the absence (-) or presence (+) of pyocyanin. The production of ROS was evaluated by flowcytometry. Each symbol represents the value for one individual. Statistical differences between the MFI obtained in the absence and the presence of pyocyanin at each time point were assessed by Wilcoxon matched-pairs signed rank test. ToD = Time of Diagnosis; EoT = End of Treatment; 3m = 3 months post EoT; 6-12m = 6–12 months post EoT. ns = not significant. (TIFF)

## Acknowledgments

We are grateful to the staff of the Leishmaniasis Research and Treatment Centre for their support and DNDi for supporting the VL treatment service at the University of Gondar.

## Author Contributions

**Conceptualization:** Yegnasew Takele, Ingrid Müller, Pascale Kropf.

**Data curation:** Yegnasew Takele, Emebet Adem, Tadele Mulaw, James Anthony Cotton, Pascale Kropf.

**Formal analysis:** Yegnasew Takele, Emebet Adem, Ingrid Müller, James Anthony Cotton, Pascale Kropf.

**Funding acquisition:** Yegnasew Takele, Ingrid Müller, James Anthony Cotton, Pascale Kropf.

**Investigation:** Yegnasew Takele, Emebet Adem, Tadele Mulaw, James Anthony Cotton, Pascale Kropf.

**Project administration:** Pascale Kropf.

**Supervision:** Ingrid Müller, James Anthony Cotton, Pascale Kropf.

**Writing – original draft:** Yegnasew Takele, James Anthony Cotton, Pascale Kropf.

**Writing – review & editing:** Yegnasew Takele, Emebet Adem, Tadele Mulaw, Ingrid Müller, James Anthony Cotton, Pascale Kropf.

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
