## [Decision Letter · Decision Letter 0]

20 Apr 2022

Dear Dr. Kropf,

Thank you very much for submitting your manuscript "Following successful anti-leishmanial treatment, neutrophil counts, CD10 expression and phagocytic capacity remain reduced in visceral leishmaniasis patients co-infected with HIV" for consideration at PLOS Neglected Tropical Diseases. As with all papers reviewed by the journal, your manuscript was reviewed by members of the editorial board and by several independent reviewers. In light of the reviews (below this email), we would like to invite the resubmission of a significantly-revised version that takes into account the reviewers' comments. 

The manuscript is well written and the experiments well carried on. However, the referees pointed out some changes that can improve the manuscript and it will be important to observe the results and tabels as well as the statistical analysis.

We cannot make any decision about publication until we have seen the revised manuscript and your response to the reviewers' comments. Your revised manuscript is also likely to be sent to reviewers for further evaluation.

Sincerely,

Claudia Ida Brodskyn

Associate Editor

Charles Jaffe

Deputy Editor

The manuscript is well written and the experiments well carried on. However, the referees pointed out some changes that can improve the manuscript and it will be important to observe the results and tabels as well as the statistical analysis.

Reviewer's Responses to Questions

**Key Review Criteria Required for Acceptance?**

**Methods**

-Are the objectives of the study clearly articulated with a clear testable hypothesis stated?

-Is the study design appropriate to address the stated objectives?

-Is the population clearly described and appropriate for the hypothesis being tested?

-Is the sample size sufficient to ensure adequate power to address the hypothesis being tested?

-Were correct statistical analysis used to support conclusions?

-Are there concerns about ethical or regulatory requirements being met?

Reviewer #1: (No Response)

Reviewer #2: The results have to be expressed as means±standard deviation that measures the dispersion of individual data relative to its mean. Depending on the distribution of the values, is better to employ a median±interquartile range. Then, means±standard error of means is not applicable to the data presented in this manuscript.

Reviewer #3: This is fine

**Results**

-Does the analysis presented match the analysis plan?

-Are the results clearly and completely presented?

-Are the figures (Tables, Images) of sufficient quality for clarity?

Reviewer #1: (No Response)

Reviewer #2: 1) The comparison with controls groups only is not adequate once it does not consider the impact of HIV infection on the VL/HIV results. On the other hand, as the two groups of patients were follow-up until the 6–12 months after therapy, it was possible to evaluate the impact of the treatment on the neutrophil counts and functionality. Then, the comparisons between VL and VL/HIV will better evaluate the impact of HIV infection on the results presented by VL.

2) The format of the tables has to be revised. Indeed, the titles are not adequate as they do not explain the contents of the results presented. The control group should be removed - see comments above.

Reviewer #3: The results are clearly presented. Figure legend may be clearer (see details in general comments)

**Conclusions**

-Are the conclusions supported by the data presented?

-Are the limitations of analysis clearly described?

-Do the authors discuss how these data can be helpful to advance our understanding of the topic under study?

-Is public health relevance addressed?

Reviewer #1: (No Response)

Reviewer #2: No

Reviewer #3: The conclusions are supported by the data, the detail of the mechanisms involved are not investigated but discussed in the discussion section.

**Editorial and Data Presentation Modifications?**

Reviewer #1: (No Response)

Reviewer #2: No applicable

Reviewer #3: The abstract would be clearer if rewritten, not starting by "As compared to..."

**Summary and General Comments**

Reviewer #1: This brief paper provides new information on various aspects of neutrophil biology in VL patients in comparison to patients with VL/HIV co-infection, the later representing a sizable part of the VL patient population in Ethiopia. The impressive size of the patient cohorts studied is a major strength of these studies. The paper is well written and the data appropriately presented. 

 While neutrophil dysfunction in VL patients, reflected by their low numbers, activation status and effector functions, were described by the same group previously, the paper extends the time of followup post-treatment to show the recovery of most of these measures. By contrast, the neutrophils from VL/HIV patients failed to be restored in their numbers, phagocytic capacity, and expression of CD10, which is an endopeptidase whose expression is associated with activated neutrophils. The authors conclude that the failure to restore neutrophil numbers and effector functions following treatment in VL/HIV patients suggests that neutrophils play a key role in the poor prognosis of these patients. The problem with this argument, however, is that the failure to restore neutrophil numbers and CD10 expression were only observed in the VL/HIV patients who relapsed their VL, not the non-relapsed patients, and thus may be secondary to their VL relapse. In this respect, these findings are not novel, since the authors have already described neutrophil impairment in patients with primary VL. The one parameter that was not restored in the non-relapsed VL/HIV patients was phagocytic function. This distinction should have been more fully discussed.

Reviewer #2: Previous results from the group (Yizengaw et al 2016) have already shown that neutrophils activity was altered in active VL patients. In this manuscript, the authors extended the studies to VL/HIV patients and compared them to other VL alone and healthy controls. The patients were follow-up at 4 different time points from the active VL up to 6-12 months after the end of anti-leishmanial therapy. The majority of VL/HIV patients were under antiretroviral therapy. Neutrophil studies were based on cell counts in blood, expression of molecules related to activation (CD63 and CD62L), neutrophil segmentation (CD10) and neutrophil function (arginase, ROS, phagocytosis of particles). In comparison to VL alone patients, VL-HIV patients did not recover neutrophils counts and maintained reduced phagocytic ability at 6-12 after therapy. The subject is interesting and the study was well conducted, however, as presented here the results are not suitable to be published.

Major comments:

1) The results related to the inability of VL/HIV patients to recover neutrophils counts VL are interesting but are predictable due to the impaired bone marrow haematopoiesis. It can render immature neutrophils as suggested by lower expression of CD10.

2) The authors suggested that “the poor prognosis of VL/HIV patients is likely to be further worsened by the inability of neutrophils from VL/HIV patients to phagocytose bacterial particles”. However, there was no statistical difference when VL/HIV relapsed and no relapsed patients were compared.

Reviewer #3: In this manuscript the authors have compared the number, activation status and effector functions of blood neutrophils between VL/HIV and VL patients at the time of VL diagnosis and after antileishmanial treatment. They have observed that both cohorts of patients experienced neutropenia, increased activation and impaired effector functions compared to neutrophils from healthy controls. However, while antileishmanial treatment restored the parameters assessed to basal levels in VL patients, neutrophils counts and CD10 expression remained significantly lower in VL/HIV patients. This work follows an Ethiopian cohort over a year, providing interesting insights into the importance of the persisting impaired neutrophil counts and response observed in VL/HIV patients after anti-leishmanial treatment. 

Major comments:

- The first major finding of this study is the maintenance of low neutrophil counts in VL/HIV patients. The authors should better discuss the possible mechanisms linked with this process.

- The second important finding is the low expression of CD10 in neutrophils of VL/HIV patients, that persist after treatment. The peptidase activity of CD10 may or not promote inflammation depending on many factors including the presence of the substrate. This should be clarified in the discussion. In line with their hypothesis, have the authors assess if markers of inflammation are more elevated in the blood of VL/HIV patients compared to that of VL patients. It seems as they performed this analysis in recent publication (supplementary figure) using the same cohort. This should be included in the discussion. 

- The role of ROS is said to be detrimental however, ROS has also been shown to destroy parasites, and be beneficial in VL, this duality should be better discussed. 

- The authors have introduced the role of neutrophils in general and in VL. The authors should also include the effect of HIV on neutrophils in the discussion of their results.

- Please comment on the inclusion of non-endemic healthy controls in the study, and the possible differences that could have been observed using endemic healthy controls. 

Minor comments:

- The authors may consider not to start the abstract with “As compared”… 

- The authors have previously shown that 78.1 % of VL/HIV patients relapse over 3 years. In the present study, approximately 30% of VL/HIV patients relapse over 1 year. Are more relapses observed after the first year following infection?

- The titles of the figures and tables should be more informative. Figure S1 does not have a title.

- Please pay attention to the use of abbreviations, for instance, the term ROS is introduced three times: in the introduction, in the methods and in the discussion. 

- Some sentences are very long, separated by ; for instance p. 19 . the sentence starting with CD63, is 8 lines long. Please shorten . 

- Please update the reference 24

PLOS authors have the option to publish the peer review history of their article (what does this mean?). If published, this will include your full peer review and any attached files.

Reviewer #1: No

Reviewer #2: Yes: Alda Maria Da-Cruz

Reviewer #3: No
---

## [Editor Report · Decision Letter 1]

20 Jul 2022

Dear Dr. Kropf,

We are pleased to inform you that your manuscript 'Following successful anti-leishmanial treatment, neutrophil counts, CD10 expression and phagocytic capacity remain reduced in visceral leishmaniasis patients co-infected with HIV' has been provisionally accepted for publication in PLOS Neglected Tropical Diseases.

Best regards,

Claudia Ida Brodskyn

Academic Editor

Charles Jaffe

Section Editor

The authors answered all the question raised by the referees and the manuscript improved its format and presentation of data.

---

## [Editor Report · Acceptance letter]

3 Aug 2022

Dear Dr. Kropf,

We are delighted to inform you that your manuscript, "Following successful anti-leishmanial treatment, neutrophil counts, CD10 expression and phagocytic capacity remain reduced in visceral leishmaniasis patients co-infected with HIV," has been formally accepted for publication in PLOS Neglected Tropical Diseases.

Best regards,

Shaden Kamhawi

co-Editor-in-Chief

Paul Brindley

co-Editor-in-Chief
